# Garcienone, a Novel Compound Involved in Allelopathic Activity of *Garcinia Xanthochymus* Hook

**DOI:** 10.3390/plants8090301

**Published:** 2019-08-24

**Authors:** Md. Mahfuzur Rob, Arihiro Iwasaki, Ryota Suzuki, Kiyotake Suenaga, Hisashi Kato-Noguchi

**Affiliations:** 1Department of Applied Biological Science, Faculty of Agriculture, Kagawa University, Miki 761-0795, Japan; 2The United Graduate School of Agricultural Sciences, Ehime University, 3-5-7 Tarumi, Matsuyama 790-8566, Japan; 3Department of Chemistry, Faculty of Science and Technology, Keio University, 3-14-1 Hiyoshi, Kohoku, Yokohama 223-8522, Japan; 4Kawasaki Refinery, JXTG Nippon Oil & Energy Co., 7-1, Ukishima-cho, Kawasaki-ku, Kawasaki-shi 210-8523, Japan

**Keywords:** allelopathy, *Garcinia xanthochymus*, growth inhibition, allelopathic substances, garcienone

## Abstract

Plants are sources of diversified allelopathic substances that can be investigated for use in eco-friendly and efficient herbicides. An aqueous methanol extract from the leaves of *Garcinia xanthochymus* exhibited strong inhibitory activity against barnyard grass (*Echinochloa crus-galli* (L.) P. Beauv.), foxtail fescue (*Vulpia myuros* (L.) C.C.), alfalfa (*Medicago sativa* L.), and cress (*Lepidium sativum* L.), and appears to be a promising source of allelopathic substances. Hence, bio-activity guided purification of the extract through a series of column chromatography steps yielded a novel compound assigned as garcienone ((*R, E*)-5-hydroxy-5-((6S, 9S)-6-methyl-9-(prop-13-en-10-yl) tetrahydrofuran-6-yl) pent-3-en-2-one). Garcienone significantly inhibited the growth of cress at a concentration of 10 μM. The concentrations resulting in 50% growth inhibition (*I*_50_) of cress roots and shoots were 120.5 and 156.3 μM, respectively. This report is the first to isolate and identify garcienone and to determine its allelopathic potential.

## 1. Introduction

Agricultural output largely depends on the efficient management of different pests, especially weeds. Weeds are considered the most harmful pest for crops, limiting their growth and development by competing for resources [1,2]. Weeds also harbor different insects and pathogens [3,4]. Many approaches concentrate on how to manage weed infestations in an effective way. Even though synthetic herbicides are the most efficient method of weed control, they have enormous hazardous effects on human health and the environment [5,6,7]. In addition, non-judicious and repeated use of synthetic herbicides has resulted in resistant weed biotypes [8,9]. Therefore, there is a crucial need to develop ecofriendly bio-herbicides. Natural products with allelopathic activities (allelochemicals) are profound source of structural diversity with unique modes of action [10,11]. When allelochemicals are released into the environment, they can suppress the growth of adjacent plants by manipulating different physiological events such as cell division and extension, protein synthesis, nutrient uptake, and membrane permeability; this phenomenon is called allelopathy [12,13]. Bio-herbicides containing allelochemicals with novel modes of action could offer numerous advantages including the prevention of herbicide-resistant weed biotypes and the maintenance of the ecological balance [14,15,16]. It is assumed that there are about 1.4 million allelopathic compounds in plants, and of those, only 3% have been examined [17]. Therefore, there is high demand to investigate promising allelochemicals that could be successfully used to develop bio-herbicides.

*Garcinia xanthochymus* Hook. f. ex T. Anderson, commonly known as false mangosteen, is a medium-sized tree belonging to the family Clusiaceae [18]. It grows up to 15–18 m tall, and its fruit are 2–3.5 cm wide and globular in shape with a prominent beak. The leaves are 25–40 cm long, 4–10 cm broad, and leathery. It is widely distributed in Southeast Asia including Bangladesh, India, China, and Myanmar [19]. The fruit and leaves of this plant are edible. Its acidic fruit are also used in jams, curries, beverages, and making preservatives [20,21]. It has been widely used in folk medicine since ancient times to treat different diseases like bilious conditions, diarrhea, and dysentery [22]. Previous studies showed that this plant possesses numerous phytochemicals with antioxidant, cytotoxic, antibacterial, and anti-inflammatory activities [23]. Because of the different bioactivities of *G. xanthochymus*, it is believed that it may also possess allelopathic activity. However, to date, there have been no reports on the allelopathic activity of *G. xanthochymus* and its potential allelopathic substances. Therefore, this study was conducted to investigate the allelopathic activity of *G. xanthochymus* and to isolate and identify the causative allelopathic substances.

## 2. Results

### 2.1. Allelopathic Activity of Garcinia Xanthochymus Leaf Extracts

The aqueous methanol extracts of *G. xanthochymus* leaves affected the growth of all the tested plant species (Figure 1 and Figure 2). At a concentration 0.1 g dry weight (DW) equivalent extract/mL, the inhibitory effects on the shoot growth of alfalfa, foxtail fescue, and barnyard grass were 99.33%, 90.30%, and 60.06% compared to control, respectively, while the cress shoots were completely inhibited. At the same concentration, the root growth of cress, alfalfa, barnyard grass, and foxtail fescue was inhibited by 98.8%, 98.40%, 95.7%, and 89.70%, respectively. In addition, at a concentration of 0.3 g DW equivalent extract/mL, the shoot and root growth of all the test plants were completely inhibited, except the shoot growth of barnyard grass and the root growth of foxtail fescue, whose growth inhibition was 99.05% and 95.90%, respectively, compare to that of control. Concentrations required for 50% (*I*_50_) shoot and root growth inhibition of all the test plants ranged from 7.1 to 65.3 and 4.7 to 14.2 mg DW equivalent extract/mL, respectively (Table 1).

### 2.2. Structure Determination of Active Substance

The ethyl acetate and aqueous fractions of the plant extracts showed concentration-dependent inhibitory activity against cress (Figure 3 and Figure 4). At the concentration of 0.3 g DW equivalent extract/mL, the ethyl acetate fraction inhibited the shoot and root growth of cress to 1.51% and 2.0% of control, respectively, while the aqueous fraction inhibited the shoot and root growth to 11.49% and 9.68%, respectively. Accordingly, the ethyl acetate fraction was chosen for further purification due to the greater allelopathic potential of that fraction.

This ethyl acetate fraction was fractionated over a silica gel column, resulting in nine fractions. All the fractions were tested for allelopathic activity against cress. Of the fractions, the highest activity was with fraction six (eluted with ethyl acetate/*n*-hexane (70:30)). That active fraction was then subjected to cress inhibitory activity-guided separation using Sephadex LH-20, reverse-phase C_18_ Sep-Pak cartridges, and HPLC (Figure 5). At each step, the most active fraction was further purified, leading to the isolation of garcienone ((*R*, *E*)-5-hydroxy-5-((6S, 9S)-6-methyl-9-(prop-13-en-10-yl) tetrahydrofuran-6-yl) pent-3-en-2-one) (Figure 6). Garcienone was obtained as a colorless oil. The molecular formula of garcienone was found to be C_13_H_20_O_3_ using HRESIMS. The NMR spectral data are summarized in Table 2.

### 2.3. Biological Activity of Garcienone

Garcienone isolated from *G. xanthochymus* was tested for biological activity against cress. The shoot and root growth of cress was significantly affected by the compound at concentrations higher than 30 and 10 μM, respectively (Figure 7). At a concentration of 1000 μM, the shoot and root growth of cress was inhibited by 89.09 and 91.89% of control, respectively. The inhibitory activity of the compound corresponded with the concentration. The *I*_50_ values of the compound for the shoot and root growth of cress were 156.32 and 120.56 μM, respectively.

## 3. Discussion

The aqueous methanol extracts of *G. xanthochymus* markedly inhibited the seedling growth of alfalfa, cress, barnyard grass, and foxtail fescue (Figure 1), and the magnitude of inhibition correlated with the extract concentration. In addition, the *I*_50_ values of the test plants varied, indicating the inhibitory activity was species specific (Table 1). Such concentration- and species-dependent inhibitory activity of several plant extracts was also documented in other studies [24,25,26]. Thus, the growth inhibitory activity of the *G. xanthochymus* extracts on the tested plants indicates that the extracts may contain allelopathic substances. In addition, it was notable that the extracts of *G. xanthochymus* inhibited diverse plant species including monocotyledonous and dicotyledonous plants, suggesting this plant could be used to control the growth of a broad range of target plant species.

In the ^1^H NMR spectrum (Appendix A), a pair of methine signals (*δ*_H_ 6.76 and 6.45) suggested the presence of an α, β-unsaturated carbonyl group. In addition, three singlet methyl signals at 2.28, 1.73, and 1.26 ppm indicated the presence of acetyl, vinyl methyl, and aliphatic methyl groups, respectively. The ^13^C NMR spectrum (Appendix A) revealed a shielded signal at 198.3 ppm corresponding to an unsaturated ketone group. Furthermore, three oxygenated carbons were expected based on three signals at 85.6, 84.7, and 76.3 ppm. In addition to these assumptions, we analyzed detailed 2D NMR spectra (Appendix A) and clarified the planar structure of garcienone as shown in Figure 8. The geometry of the olefin at C3 was determined to be *E* based on the large coupling constant between H-3 and H-4 (16.2 Hz). Although the presence of an ethereal ring was predicted based on its molecular formula, we could not observe any transannular HMBC correlations such as H6/C9 or H9/C6. However, the 5-OH proton signal was observed as a doublet signal coupled with H-5 (1.1 Hz) in the ^1^H NMR spectrum. Therefore, we concluded that garcienone possessed a five-membered ethereal ring, not a six-membered ring, and determined the planar structure of garcienone as shown in Figure 8.

We then clarified the relative configuration of garcienone and it possessed three stereocenters at C5, C6, and C9. Two NOE correlations, H12/H11 and H12/H13a, indicated that a methyl group and an isopropenyl group were located in the same face on the five-membered ring. Therefore, we determined the relative configuration of C6 and C9 to be 6*S**, 9*S** as shown in Figure 9.

However, we could not obtain any useful information regarding the relative configuration at C5 from NMR experiments. Recently, computational chemistry based on density functional theory (DFT) has been applied to clarify stereochemistries of natural products, and several methodologies have been developed [27,28,29]. One accepted method is the DP4+ method developed by Grimblat and others [30], and we used this method to elucidate the relative stereochemistry of garcienone (Appendix A). First, we calculated the theoretical chemical shifts of four possible diastereomers, 5*S*,6*S*,9*S*, 5*R*,6*S*,9*S*, 5*S*,6*R*,9*S*, and 5*R*,6*R*,9*S*, according to a previous study report [31] (for details, see Materials and Methods). The calculated chemical shifts were subjected to DP4+ analysis along with those of natural garcienone [30]. As a result, the diastereomer 5*R**,6*S**,9*S** exhibited a significantly higher DP4+ probability score (100.00%). Therefore, we determined the relative configuration of garcienone to be 5*R**,6*S**,9*S**.

Finally, we elucidated the absolute configuration of garcienone. Only a limited amount of garcienone was available, so we could not carry out any derivatization reactions such as a modified Mosher method. Therefore, we compared the ECD spectrum of garcienone with the calculated ECD data of two possible enantiomers: 5*R*,6*S*,9*S* and 5*S*,6*R*,9*R* (Figure 9, and Appendix A). For details of the calculation, see Materials and methods. Although the signal-to-noise ratio of the ECD data of natural garcienone was somewhat poor, the distribution of the curves suggested that garcienone possessed the 5*R*,6*S*,9*S* configuration. Thus, we propose the absolute configuration of garcienone to be 5*R*,6*S*,9*S* and garcienone was assigned as (*R*, *E*)-5-hydroxy-5-((6*S*, 9*S*)-6-methyl-9-(prop-13-en-10-yl) tetrahydrofuran-6-yl) pent-3-en-2-one.

The identified compound, garcienone, markedly inhibited the shoot and root length of cress. The inhibitory activity varied with compound concentration, in which the higher the concentration, the more inhibition occurred in the growth of the test plants. Previous reports also documented concentration-dependent inhibitory activity of allelopathic substances [32,33]. Moreover, based on the *I*_50_ values, the inhibitory activity of garcienone was greater against the roots, compared with the shoots. Previous reports also confirmed that root growth is more sensitive to allelochemicals than shoot growth [34,35]. Indeed, the identification of allelopathic substances from *G. xanthochymus* may lead to developing new bioherbicides by using allelopathic potency of the compound and thus can play a pivotal role in weed management in a more environmentally sound way.

## 4. Materials and Methods

### 4.1. General Experimental Procedures

Optical rotations were measured using a JASCO DIP-1000 polarimeter. UV spectra were recorded on a JASCO V730-BIO Spectrophotometer. CD spectra were measured using a JASCO J-720 W spectropolarimeter. IR spectra were recorded on a JASCO FT/IR-4200 spectrometer. All NMR data were recorded on a JEOL JNM-ECX400 spectrometer for ^1^H (400 MHz) and ^13^C (100 MHz). ^1^H NMR chemical shifts (referenced to residual CHCl_3_ observed at *δ*_H_ 7.26) were assigned using a combination of data from COSY and HMQC experiments. Similarly, ^13^C NMR chemical shifts (referenced to CDCl_3_ observed at *δ*_C_ 77.16) were assigned based on HMBC and HMQC experiments. HRESIMS spectra were obtained on an LCT Premier XE time-of-flight (TOF) mass spectrometer (LCT premier XE; Waters). Reverse-phase HPLC was performed (500 × 10 mm I.D. ODS AQ-325; YMC Co., Ltd., Kyoto, Japan), eluted at a flow rate of 1.5 mL/min with 45% (*v*/*v*) aqueous methanol, and detected at wavelength 220 nm.

### 4.2. Plant Materials

Mature leaves of *Garcinia xanthochymus* Hook. f. ex T. Anderson were collected from Netrokona Sadar (24.8750° N 90.7333° E), Netrokona, Bangladesh in July 2017. The plant species was identified by crop botanist Dr. Masudur Rahman, Associate Professor, Department of Crop Botany and Tea Technology. A voucher specimen (SAUCB 19127) was submitted to the Crop Botany Herbarium at Sylhet Agricultural University, Sylhet, Bangladesh. The collected leaves were well cleaned and shed dried. The dried samples were then ground into fine powder using a grinding machine and stored at 2 °C until further use. For the bioassay experiment, seeds of two monocotyledonous species [barnyard grass (*Echinochloa crus-galli* (L.) P. Beauv.) and foxtail fescue (*Vulpia myuros* (L.) C.C. Gmel.)] and two dicotyledonous species [alfalfa (*Medicago sativa* L.) and cress (*Lepidium sativum* L.)] were used as test plants. Cress and alfalfa were used for their well-known growth behaviors, while barnyard grass and foxtail fescue were used because of their common distribution in fields around the world.

### 4.3. Extraction and Bioassay

Powdered leaves of *Garcinia xanthochymus* (100 g) were extracted with 500 mL of 70% (*v*/*v*) aqueous methanol. After 48 h, the solution was filtered using one layer of filter paper (No. 2, 125 mm; Toyo Ltd., Tokyo, Japan). The residue was re-extracted with 500 mL of cold methanol for 24 h and filtered. Two filtrates were then combined and concentrated in a rotary evaporator at 40 °C until dry. The resulting crude extract of *G. xanthochymus* was dissolved in 250 mL methanol to prepare six assay concentrations (0.001, 0.003, 0.01, 0.03, 0.1, and 0.3 g dry weight (DW) equivalent extract/mL). To obtain those concentrations, aliquots of methanol extract (1.5, 4.5, 15, 45, 150, and 450 µL, respectively) were added over the filter paper (No. 2, 28 mm; Toyo) in 28 mm Petri dishes. After drying the methanol in a fume hood, the filter papers in the Petri dishes were soaked with 0.6 mL of 0.05% (*v*/*v*) aqueous solution of polyoxyethylene sorbitan monolaurate (Tween 20; Nacalai Tesque, Inc., Kyoto, Japan). Ten seeds of alfalfa and cress, and 10 seedlings of barnyard grass and foxtail fescue (germinated in darkness at 25 °C for 72 and 48 h, respectively, after overnight soaking in distilled water in Petri dishes in each case) were added to separate Petri dishes. Simultaneously, control seeds or seedlings received 0.6 mL of 0.05% (*v*/*v*) aqueous solution of Tween 20. All the dishes were then covered with polyethylene film and aluminum foil and kept in a growth chamber (25 °C, darkness). After 48 h of incubation, the length of the roots and shoots of these seedlings was measured and compared with the control seedlings. The inhibition percentage was calculated using following equation:
Inhibition(%)=[1−(length of treated shoot or root÷length of control shoot or root)]×100

### 4.4. Purification of the Active Substance

The shed-dried and powdered leaves (2 kg) of *Garcinia xanthochymus* Hook were extracted as described above. The extracts were then evaporated using a rotary evaporator (40 °C) to yield an aqueous residue. The aqueous residue was fixed to pH 7.0 using 1 M phosphate buffer and partitioned three times with the same amount of ethyl acetate. The aqueous and ethyl acetate fractions were subjected to a growth bioassay with cress as described above. The ethyl acetate fraction was dried using a rotary evaporator after overnight soaking with anhydrous Na_2_SO_4_. The dried crude mass was then subjected to silica gel column chromatography (60 g of silica gel 60, spherical, 70–230 mesh; Nacalai Tesque, Inc.), eluted stepwise with *n*-hexane (150 mL per step) containing increasing amounts of ethyl acetate (10% per step, *v*/*v*) from 20 to 80%, ethyl acetate (150 mL), and methanol (300 mL). The highest biological activity was obtained by elution with 70% ethyl acetate in the *n*-hexane fraction checked with the previously described cress bioassay. The active fraction was then purified using a column of Sephadex LH-20 (GE Healthcare Bio-Sciences AB, SE-751 84, Uppsala, Sweden) and eluted with 20, 30, 40, 50, 60, and 80% (*v*/*v*) aqueous methanol (150 mL per step) and methanol (300 mL). The most active fraction was obtained by elution with 40% aqueous methanol and evaporated to dryness. The residue was suspended in 20% (*v/v*) aqueous methanol and loaded onto a reverse-phase C_18_ cartridge. The cartridge was eluted with 20, 30, 40, 50, 60, and 80% (*v/v*) aqueous methanol (15 mL per step) and methanol (30 mL per step). The active fraction was obtained by elution with 50% aqueous methanol and its content was purified using reverse-phase HPLC (500 × 10 mm I.D. ODS AQ-325; YMC Ltd., Kyoto, Japan) at a flow rate of 1.5 mL/min with 45% aqueous methanol and detected at wavelength 220 nm and 40 °C. The inhibitory peak was found at retention time 210–240 min and again purified using reverse-phase HPLC (4.6 × 250 mm I.D., S-5 µm, Inertsil^®^ ODS-3; GL Science Inc., Tokyo, Japan) at a flow rate of 0.8 mL/min with 30% aqueous methanol (wavelength: 254 nm; oven temperature: 40 °C, and retention time: 90–120 min). The structural elucidation of the substance was achieved using HRESIMS, IR, ^1^H-NMR (400 MHz, CDCl_3_), HMBC, NOESY, ^13^C-NMR spectra (100 MHz, CDCl_3_), and optical rotation.

### 4.5. Characterization of the Compound

#### 4.5.1. NMR Chemical Shifts Calculation Methods

The conformational search for 5*S*,6*S*,9*S*-**1**, 5*R*,6*S*,9*S*-**1**, 5*S*,6*R*,9*S*-**1**, and 5*R*,6*R*,9*S*-**1** was carried out using the OPLS 2005 force field and torsional sampling of the Macro Model program [31,36]. An energy window cutoff of 5.0 kcal/mol was used during the conformational search. Redundant conformers were eliminated using an RMSD cutoff of 1.0 Å. All conformers were subjected to geometry optimization using the Gaussian 16 package [37], in the gas phase at the B3LYP/6-31G* level. The resulting conformers within 2 kcal/mol of each global minimum were proceeded to gauge-invariant atomic orbital (GIAO) shielding constant calculations [38], at the mPW1PW91/6-31+G** level with PCM in chloroform [29,39]. The calculated NMR properties were averaged based upon their respective Boltzmann populations, and the resulting data were used for calculations of DP4+ probability analysis using an Excel sheet [30].

#### 4.5.2. ECD Calculation Methods

The conformational search for 5*R*,6*S*,9*S*-**1** was carried out using the OPLS 2005 force field and torsional sampling of the Macro Model program [36]. An energy window cutoff of 5.0 kcal/mol was used during the conformational search. Redundant conformers were eliminated using an RMSD cutoff of 1.0 Å. The resulting sixteen lowest-energy conformers were further optimized at the hybrid density-functional B3LYP level of theory using the 6-31G* basis set with PCM, in MeOH using the Gaussian 16 package [37,40]. The optimized conformers that showed >1% Boltzmann population were used for ECD calculations using TDDFT [41,42], at the B3LYP/6-31G* level with PCM in MeOH. ECD curves were generated with Spec Dis software using a half-bandwidth of 0.16 eV [40,43]. The relative populations of each conformer were calculated based on the Boltzmann weighting factor at 298 K. The ECD spectrum of the enantiomer, 5*S*,6*R*,9*R*-**1**, was generated with Spec Dis [43].

### 4.6. Spectral Data

Garcienone. Colorless oil; [α]_D_^27^ = +36 (*c* 0.02, CH_3_OH); UV (MeOH) *λ*_max_ (log *ε*) 227 (4.15) nm; ECD (MeOH) 210 (Δ*ε* −280), 234 (Δ*ε* +270) nm; IR (neat) 3447, 2969, 2925, 2852, 1675, 1647, 1456, 1374, 1257 cm^−1^; HRESIMS *m*/*z* 225.1477 [M + H]^+^ (calcd for C_13_H_21_O_3_, 225.1491, Δ = −1.4 mmu); ^1^H NMR (400 MHz, CDCl_3_) δ_H_ 6.76 (dd, *J* = 16.2, 4.8 Hz, 1 H, H4), 6.45 (dd, *J* = 16.2, 1.8 Hz, 1 H, H3), 5.03 (s, 1 H, H13a), 4.84 (s, 1 H, H13b), 4.39 (dd, *J* = 10.7, 5.7 Hz, 1 H, H9), 4.30 (m, 1 H, H5), 2.65 (d, *J* = 1.1 Hz, 1 H, 5-OH), 2.28 (s, 3 H, H1), 2.09 (m, 1 H, H7a), 2.06 (m, 1 H, H8a), 1.82 (m, 1 H, H8b), 1.73 (s, 3 H, H11), 1.53 (m, 1 H, H7b), 1.26 (s, 3 H, H12); ^13^C NMR (100 MHz, CDCl_3_) δ_C_ 198.3 (C2), 145.4 (C10), 143.8 (C4), 130.9 (C3), 110.9 (C13), 85.6 (C6), 84.7 (C9), 76.3 (C5), 31.9 (C7), 31.3 (C8), 28.1 (C1), 24.6 (C12), 17.9 (C11).

### 4.7. Bioassay of the Isolated Compound

The isolated compound was dissolved in 1 mL methanol and applied to filter paper placed in 28 mm Petri dishes to obtain the final assay concentrations of 1, 3, 10, 30, 100, 300, and 1000 μM. Before moistening with 0.6 mL of 0.05% (*v*/*v*) aqueous Tween 20 solution, the filter paper was completely dried. Ten seeds of cress were arranged in each Petri dish and kept in a growth chamber under constant darkness at 25 °C. The shoot and root length of the cress seedlings were determined after 48 h of growth and compared with the control seedlings.

### 4.8. Statistics

The bioassay experiment was conducted in a completely randomized manner with three replications and repeated twice. ANOVA of all data was carried out using SPSS software, version 16.0 (SPSS Inc., Chicago, IL, USA) followed by Tukey’s test at a significance level of 0.05. The concentration causing 50% inhibition, expressed as an *I*_50_ value, was calculated using a regression equation of the concentration response curves.

## Figures and Tables

**Figure 1 plants-08-00301-f001:**
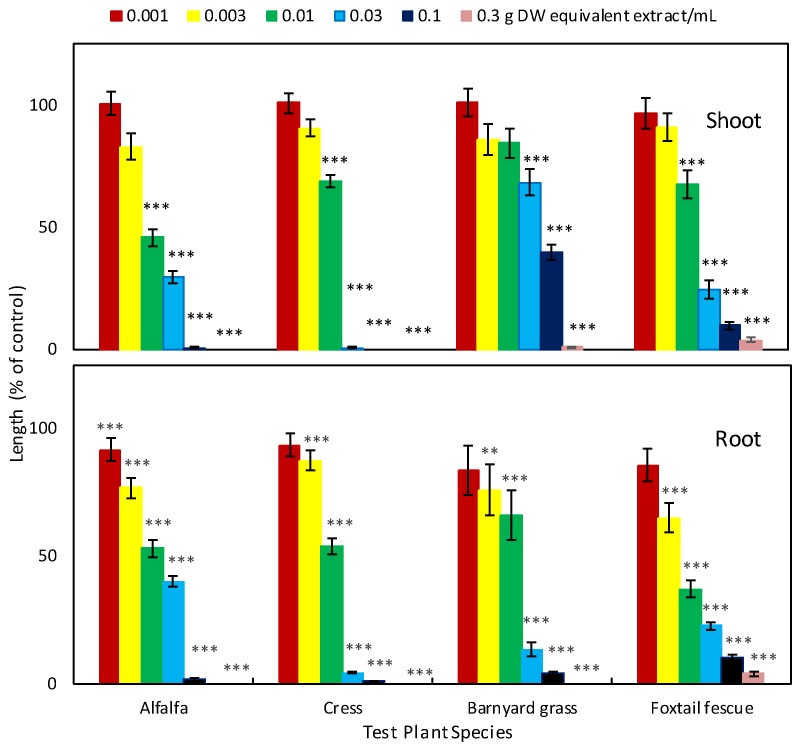
Effect of *Garcinia xanthochymus* extracts on the shoot and root growth of alfalfa, cress, barnyard grass, and foxtail fescue. All the test plant species were treated at the concentrations of 0.001, 0.003, 0.01, 0.03, 0.1, and 0.3 g dry weight equivalent extract of *G. xanthochymus*/mL. Mean ± Standard Error (SE) from two independent experiments with three replications for each treatment (number of seedlings per treatment = 10, n = 60). Each vertical bar represents standard error of the mean. Asterisks indicate significant differences between treatment and control: ** *p* ˂ 0.01 and *** *p* ˂ 0.001.

**Figure 2 plants-08-00301-f002:**
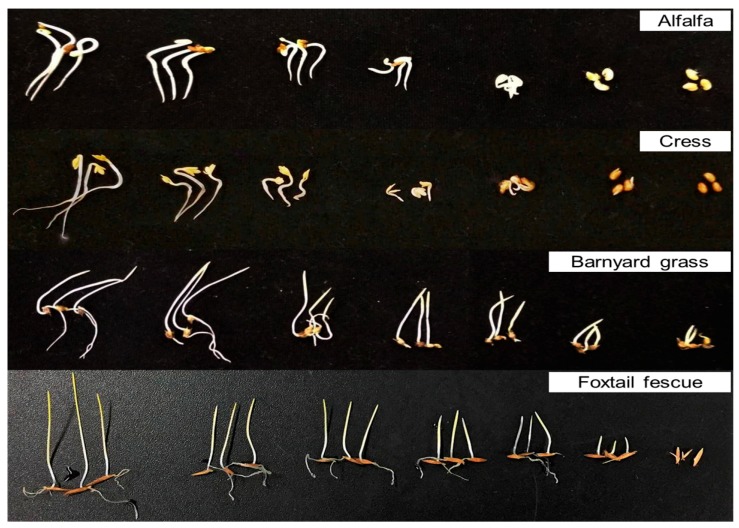
Effect of *Garcinia xanthochymus* extracts on the growth of alfalfa, cress, barnyard grass, and foxtail fescue. Treatment concentrations (from left to right in each picture): Control, 0.001, 0.003, 0.01, 0.03, 0.1, and 0.3 g dry weight (DW) equivalent extract of *G. xanthochymus*/mL.

**Figure 3 plants-08-00301-f003:**
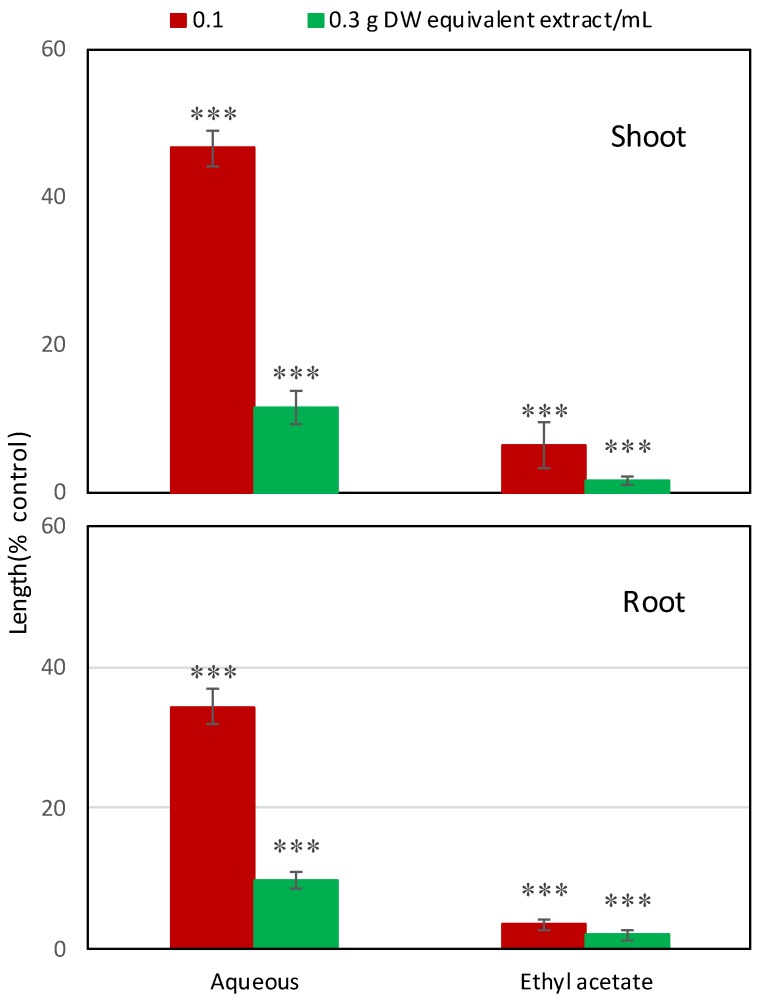
Effect of aqueous and ethyl acetate fractions on the growth of cress. The treatment concentrations were 0.1 and 0.3 g dry weight equivalent extract of *Garcinia xanthochymus*/mL. Mean ± SE from two independent experiments with 10 seeds for each treatment. Each vertical bar represents standard error of the mean. Asterisks indicate significant differences between treatment and control: *** *p* < 0.001.

**Figure 4 plants-08-00301-f004:**
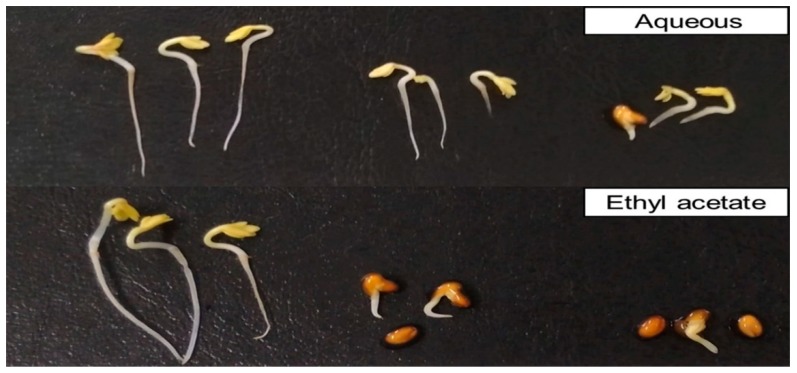
Effect of aqueous and ethyl acetate fractions on the growth of cress. Treatment concentrations (from left to right in each picture): Control, 0.1 and 0.3 g dry weight equivalent extract of *G. xanthochymus*/mL.

**Figure 5 plants-08-00301-f005:**
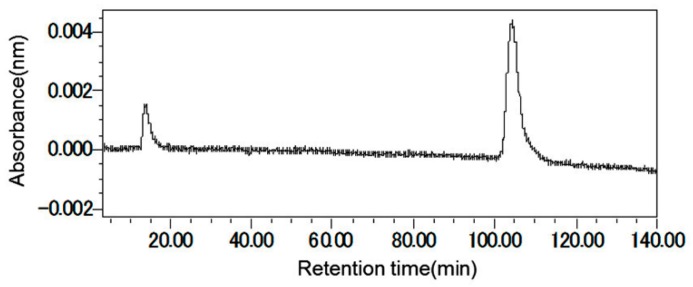
Chromatogram of the garcienone obtained from reverse phase HPLC, eluted at a flow rate of 0.8 mL/min with 30% aqueous methanol (wavelength: 254 nm; oven temperature: 40 °C, and retention time: 90–120 min).

**Figure 6 plants-08-00301-f006:**
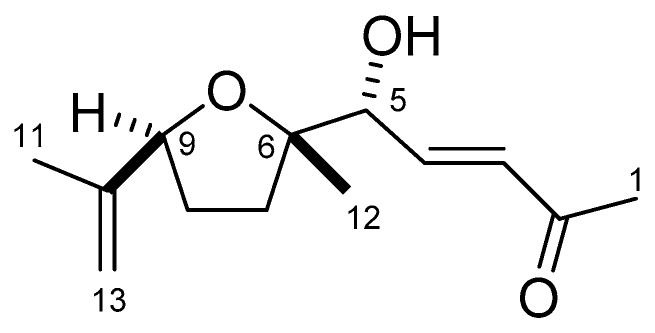
Chemical structure of garcienone ((*R*, *E*)-5-hydroxy-5-((6*S*, 9*S*)-6-methyl-9-(prop-13-en-10-yl) tetrahydrofuran-6-yl) pent-3-en-2-one).

**Figure 7 plants-08-00301-f007:**
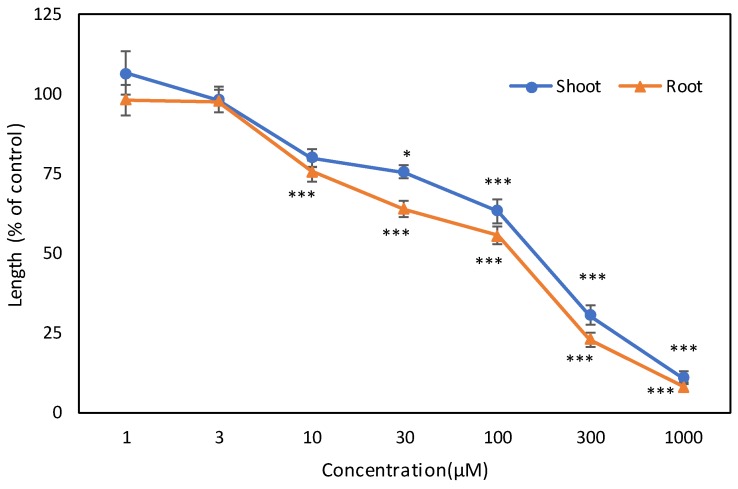
Effect of garcienone on the shoot and root growth of cress. Values represent means ± SE from three replicate Petri dishes for each treatment (n = 30). Significant differences between control and treatment are represented by * *p* < 0.05 and *** *p* < 0.001.

**Figure 8 plants-08-00301-f008:**
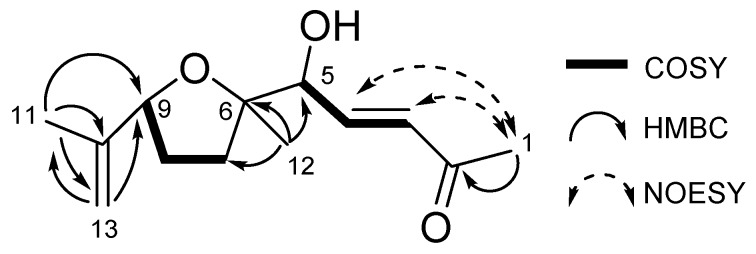
Planar structure of garcienone based on two-dimensional (2D) NMR spectra.

**Figure 9 plants-08-00301-f009:**
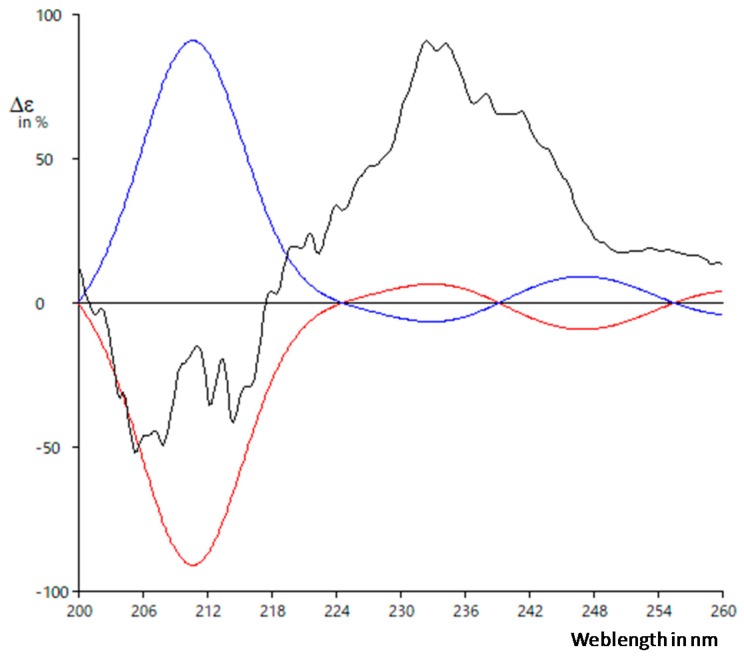
Experimental ECD spectrum of garcienone (black) and calculated ECD spectra of 5*R*,6*S*,9*S*-**1** (red) and 5*S*,6*R*,9*R*-**1** (blue).

**Table 1 plants-08-00301-t001:** The concentration required for 50% growth inhibition (*I*_50_) of the shoot and root growth of test plant species by the aqueous methanol extracts of *Garcinia xanthochymus*.

Aqueous Methanol Extracts (mg Dry Weight Equivalent Extract/mL)		
Test Plant Species	Shoot	Root
Alfalfa	8.70	13.11
Cress	13.55	10.93
Barnyard grass	65.31	13.99
Foxtail fescue	15.67	5.75

**Table 2 plants-08-00301-t002:** NMR data for garcienone in CDCl_3_.

	*δ*_C_, Type ^b^	*δ*_H_^a^ (*J* in Hz)
1	28.1, CH_3_	2.28, s
2	198.3, C	
3	130.9, CH	6.45, dd (16.2, 1.8)
4	143.8, CH	6.76, dd (16.2, 4.8)
5	76.3, CH	4.30, m
6	85.6, C	
7a	31.9, CH_2_	2.09, m
7b		1.53, m
8a	31.3, CH_2_	2.06, m
8b		1.82, m
9	84.7, CH	4.39, dd (10.7, 5.7)
10	145.4, C	
11	17.9, CH_3_	1.73, s
12	24.6, CH_3_	1.26, s
13a	110.9, CH_2_	5.03, s
13b		4.84, s
5-OH		2.65, d (1.1)

^a^ Measured at 400 MHz. ^b^ Measured at 100 MHz.

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
