# Peer review of "Garcienone, a Novel Compound Involved in Allelopathic Activity of Garcinia Xanthochymus Hook"

_plants, 2019, doi:10.3390/plants8090301_

Round 1
Reviewer 1 Report
The authors analysed the possible phytotoxic activity of Garcinia xanthochymus Hook acqueous extract and of its component Garcienone, a novel compound identified in this research.
A few issue should be addressed:
- DW equivalent extract/mL please explain DW the first time you mentioned it in the text.
- figure 1,2 ..why you use Mean ± SE and not Mean ± SD?
- figure 1 ... please adjust errors bars and asterics in the same vertical line
- why you change units for garcienone that it is express in uM and not in g/mL as the extract? how you can compare the results?
Reviewer 2 Report
Overall, this manuscript is of great interest for discovery of natural products with phytotoxic activities. The experimental plans were thoughtfully formulated. The experiments were executed systematically. However, I have minor concerns that should be addressed to improve the quality of the manuscript.
General concerns
The authors concluded that the identification of phytotoxic substances from G. xanthochymus/Garcinone may lead to the development of new bioherbicides. The experiments for growth inhibitions were conducted at 25oC in darkness using desiccated seeds or germinating seeds. It will enhance the quality of the manuscript if the authors conduct experiments in the light/dark cycles which mimic the natural growth conditions. The authors should also consider using various stages of the tested plants e.g. stages with cotyledons, true leaves. This will provide better understanding of the effects of this novel compound.
In the bioassay experiment, seeds/germinating seeds were placed on filter papers containing plant extracts/Garcinone. It would be interesting to see if the active compound(s) would be transported to the shoot (and subsequently act as shoot growth inhibitor) if only roots were submerged in the active compound(s). Or, does it mean that the active compound(s) have to make direct contact with the shoot to be able to inhibit growth. This will be vital in developing herbicides since the roots should be able to absorb herbicide from soil and transport it through the whole plant.
Phenotypic differences of the tested plants should be shown along in Figure 1 and 2.
Minor concerns
In the result part where the authors showed inhibitory effects of Garcinia xanthochymus leaf extracts, it will read more smoothly if the values were reported as percent inhibition (compared to control). For example, line 66 : “At the same concentration, the root growth of cress, alfalfa, barnyard grass, and foxtail fescue was suppressed to 1.20, 1.60, 4.30, and 10.30% of control, respectively.” Should read “At the same concentration, the inhibitory effects on root growth of cress, alfalfa, barnyard grass, and foxtail fescue were 98.8, 98.4, 95.7, and 89.7% compared to that of control, respectively.”
The structure elucidation part of this manuscript is well presented and supported in supplemental data. However, the HPLC trace that led to the isolation of the pure Garcinone should be presented here as well.
Reviewer 3 Report
This is the best paper sent to my revision in the past months, I feel I must congratulate the authors for such a fine job, very interesting and easy to read. It is well search, with appropriate references, not too many.
My only issue is with the choice of "phytotoxicity" to describe the observed activity, I think "allelopathy" would be more adequate since it is the only type of effect studied in this paper.
For the same reason, caution is advised on using expressions such as "it could be used as a source of natural herbicides" it can also show other types of toxicity.
Minor mistakes and punctuation should be revised.
Overall, a very good paper
Round 2
Reviewer 1 Report
The manuscript can be published in the current forum.